## Perspective

mental health; handloom weavers; occupational stress; healthcare access; Bangladesh

**Corresponding author:**
Md. Ashiquir Rahaman;
Email: ashiqur@du.ac.bd

# Mental health of handloom weavers in Bangladesh: A call for culturally adapted interventions

Md. Ashiquir Rahaman 

Department of Clinical Psychology, Dhaka University, Dhaka, Bangladesh

## Abstract

Bangladesh's handloom weaving industry, a vital cultural and economic asset, employs approximately one million rural workers and generates over 10 billion Bangladeshi taka (~82 million USD) annually. Despite its importance, the mental health of handloom weavers, locally known as Tatis, remains largely unexamined. This perspective article, based on a narrative review of existing literature synthesizing peer-reviewed studies, reports and policy documents on mental health in informal sectors, explores the mental health challenges faced by these workers. Using a syndemics framework, it draws on data on garment workers and the broader informal sector, which indicates heightened risks of stress, anxiety and depression resulting from long working hours, low wages and competition from mechanized looms. These risks are compounded by systemic barriers, including Bangladesh's allocation of only 0.44% of its health budget to mental health (2021), a severe shortage of professionals (0.16 psychiatrists per 100,000 people and 0.34 psychologist per 100,000 people) and pervasive cultural stigma. Additionally, musculoskeletal pain, which affects 82.4% of weavers, places a particularly heavy burden on women, who constitute half of the workforce, further exacerbating mental health vulnerabilities through syndemic interactions with poverty and gender inequities. To address this neglected crisis, the article proposes a novel intervention framework aligned with the Double Diamond design model. The framework integrates community-based mental health hubs, peer-led support networks and digital platforms tailored to Bangladesh's collectivist culture. It calls for increased funding, workplace reforms, stigma reduction campaigns and targeted research, highlighting the dual benefit of improving weavers' well-being and sustaining the long-term future of the industry.

## Impact statement

This article addresses a critical gap in global mental health research by examining the psychological well-being of weavers in Bangladesh, a population that represents the broader challenges faced by informal sector workers in low- and middle-income countries (LMICs). The article highlights significant mental health concerns, including stress, anxiety and depression, which are likely driven by occupational hazards, chronic physical pain and systemic barriers such as limited mental health funding and widespread cultural stigma. These findings are consistent with the challenges experienced by informal workers globally, particularly in LMICs, where mental health resources are limited and traditional beliefs frequently discourage individuals from seeking help. The article's proposed intervention framework, which includes community-based mental health hubs, peer-led support networks and digital platforms, presents an innovative and scalable model specifically designed to align with Bangladesh's collectivist social structure. This approach is transferable to similar LMICs' informal-sector contexts. Its successful implementation, however, is contingent on specific boundary conditions within the local setting, including the level of community digital connectivity and the existing density of cooperative organizations. Adaptation to local cultural and logistical specificities remains crucial for effective scale-up. By centering on marginalized populations, particularly women who face gender-specific stressors, the article contributes to the advancement of mental health equity. The article's emphasis on the need for further research, policy reform and multisectoral collaboration provides a practical roadmap for building sustainable mental health systems. These strategies not only support vulnerable workers but also help preserve traditional industries and promote broader economic resilience. By producing insights that are both locally grounded and globally transferable, this article supports a public mental health agenda that values inclusion, cultural sensitivity and long-term impact.





## Introduction

Bangladesh's handloom industry is a cornerstone of its cultural heritage and rural economy, employing around one million workers, predominantly in districts such as Sirajganj, Tangail,

Pabna, Narsingdi, Kushtia, Narayanganj, Dhaka, Brahmanbaria, Bogra and Sylhet, including indigenous Manipuri communities. The workforce comprises ~50% women, with an age distribution largely between 25 and 55 years (Liton et al., 2016; Roy Maulik, 2021; Rahman and Biswas, 2023). The sector contributes over 10 billion Bangladeshi taka (~82 million USD) annually through the production of iconic textiles, such as Jamdani and Muslin (Liton et al., 2016). This traditional craft, rooted in centuries of history, not only sustains livelihoods but also embodies national identity. However, beneath this cultural and economic significance lies a pressing public health issue: the mental well-being of weavers, or Tatis, remains critically understudied. Unlike the national mental health survey, which reported an 18.7% prevalence of adult mental disorders, there is no direct prevalence data on the mental health of this workforce (NIMH, 2021).

Weavers typically operate within the informal sector, where they endure grueling conditions, such as extended working hours, meager wages and growing competition from mechanized looms. These structural vulnerabilities exacerbate economic insecurity and heighten psychosocial stressors, which in turn contribute to mental health challenges. Viewed through the framework of the social determinants of health (Marmot et al., 2008), these intersecting factors create an environment in which mental health risks are amplified but systematically overlooked in both research and policy.

This perspective article, based on a narrative review of existing literature and synthesizing data from peer-reviewed studies, reports and policy documents on mental health in Bangladesh's informal sectors, seeks to highlight the mental health risks faced by weavers. It draws justified comparisons with other informal sector workers, such as garment workers, who experience similar socioeconomic stressors, including job insecurity and physical demands (Miching, 2021; Kabir et al., 2023). Systemic barriers further exacerbate the crisis, including Bangladesh's limited mental health budget, which accounts for only 0.44% of total health expenditure in the year 2021, a severe shortage of mental health professionals with just 0.16 psychiatrists per 100,000 people and persistent cultural stigma (Ashraf et al., 2022; Haque et al., 2022; WHO, 2025).

In addition to these challenges, the physical health burden among weavers is substantial. Recent evidence indicates that 82.4% of workers experience musculoskeletal pain, a condition that disproportionately affects women due to their dual responsibilities in both production and domestic labor (Jamil et al., 2022). These physical health challenges, particularly musculoskeletal pain, are associated with negative emotions and psychological distress and often intersect with psychosocial stressors, creating complex vulnerabilities that cannot be adequately understood in isolation (Crofford, 2015). Drawing on the framework of syndemics theory (Singer et al., 2017), this article emphasizes how co-occurring health burdens, both physical and mental, are shaped and amplified by broader social, economic and structural determinants. Recent qualitative work on female informal workers in Bangladesh traces migration-related stressors, workplace conditions and coping strategies that are germane to handloom communities; integrating this evidence would sharpen the gendered pathways we theorize here (Islam et al., 2025). We also adopt a syndemics lens in line with Singer and Clair (2003) to specify co-occurring burdens and their adverse interactions within harmful social contexts.

By synthesizing available empirical evidence with ecological and culturally grounded perspectives, this article proposes a context-sensitive intervention framework designed to address the intertwined health and livelihood concerns of weavers. The framework highlights the importance of developing integrated strategies that combine occupational health, mental health and social protection policies.

Furthermore, it advocates for multisectoral collaboration involving mental health practitioners, public health policymakers, development economists and nongovernmental organizations (NGOs) engaged in rural labor and cultural preservation. Such coordinated action is essential not only to safeguard the well-being of weavers but also to ensure the long-term sustainability of Bangladesh's handloom industry as both a cultural heritage and a vital source of livelihood.

## Mental health challenges: Prevalence and risk factors

### Occupational stress and mental health risks

The mental health of weavers remains largely unexplored because of the absence of direct studies. However, some investigations into physical health issues conducted in different regions of India briefly mention psychological aspects of weaving, particularly among women, such as mental stress, anger, rage and frustration, although prevalence rates remain unknown (Sharma et al., 2017; Shobana and Latha, 2020; Chinnu and Sheeba, 2021; Jeeva, 2022; Chakravarty, 2025). This perspective acknowledges the data gap and calls for targeted research to examine the prevalence and nature of mental health conditions among weavers through mixed-methods approaches (Hasan et al., 2021). Evidence from related informal sector workers offers a compelling basis for inference, as these groups share common social determinants such as economic precarity and low wages (Marmot et al., 2008; WHO, 2021b). For example, a 2023 study found that over 60% of informal workers in Bangladesh, including those involved in weaving, struggle to balance work and family life because of long working hours and excessive overtime demands, which contribute to stress, anxiety and reduced overall well-being (Rahman and Biswas, 2023). Research on garment workers, who encounter comparable socioeconomic pressures, provides further insight. One study reported that 23.5% of working women in Bangladesh, including 20.9% of garment workers, experience moderate to severe depression (Fitch et al., 2017). Similarly, another study documented high rates of mental health issues among garment workers, with 69.1% reporting stress, 66.2% anxiety, 64.5% boredom, 51.3% sleeplessness, 48.2% depression and 34.3% fear (Kabir et al., 2023). While these inferences from garment workers provide plausible hypotheses for weavers, given their overlapping socioeconomic challenges, contextual differences, such as family-based production in weaving, seasonal workloads and less formalized family-based loom settings, require caution and highlight the need for primary data specific to weaving communities. Given the overlapping challenges of low wages, job insecurity and demanding workloads, it is reasonable to infer that weavers bear a comparable psychological burden.

The informal nature of weaving further intensifies these risks. Unlike employees in the formal sector, weavers typically lack access to labor protections, health insurance and mental health services, which makes them more susceptible to exploitation and prolonged stress (Michlig, 2021). The growing threat of mechanization adds another layer of anxiety, as many weavers fear that their traditional skills may become obsolete in an industrialized textile market (Roy Maulik, 2021). Together, these occupational and existential pressures create an environment where mental health challenges are both prevalent and insufficiently addressed.

### Physical health and its impact on mental well-being

The physical demands of weaving play a crucial role in shaping mental health vulnerabilities. A 2022 cross-sectional study of 250 rural

weavers in Sirajganj district, a major weaving center in Bangladesh, found that 82.4% of weavers suffer from musculoskeletal pain, with 50% reporting lower back pain, 48.4% experiencing shoulder pain and 46.4% reporting knee pain (Jamil et al., 2022). This study provides district-level evidence but should not be used to generalize without broader surveys across multiple weaving clusters. This pain results from repetitive movements, extended periods of sitting and poor ergonomic conditions, which are common characteristics of traditional weaving. Chronic pain is a well-documented risk factor for mental health disorders, including depression and anxiety (Sheng et al., 2017). The bidirectional relationship between physical pain and mental health creates a harmful cycle, in which pain increases psychological distress, and psychological distress, in turn, heightens the perception of pain (Bair et al., 2003).

For female weavers, who comprise 50% of the workforce, this burden is especially pronounced. In rural Bangladesh, women often combine weaving with domestic responsibilities such as childcare and household management, which significantly increases their stress levels (Makhdum et al., 2024). In addition, gender-based discrimination and limited decision-making power within households further elevate their susceptibility to mental health challenges (Opanasenko et al., 2021; Islam and Akter, 2024; Jain and Pandey, 2025). This burden intersects with life-course factors, such as marital status (e.g., heightened vulnerabilities among widowed or separated women due to reduced social support) and ethnicity (e.g., among indigenous Manipuri weavers in Sylhet, where cultural marginalization compounds gender inequities). The combination of chronic physical pain, demanding work conditions and systemic gender inequity contributes to a disproportionately high mental health burden for female weavers.

## Systemic barriers and cultural stigma

### Inadequate mental health infrastructure

Bangladesh's mental health system is severely under-resourced, with only 0.44% of the national health budget allocated to mental health in the year 2021 (WHO, 2025). This figure falls well below the 5% advocated by global mental health groups for LMICs (Patel et al., 2018; WHO, 2021b; Haque et al., 2022). The mental health workforce is also insufficient, comprising only 260 psychiatrists (0.16 per 100,000 people), 700 nurses who provide mental health specialty care (0.4 per 100,000) and 565 psychologists (0.34 per 100,000) across the country (WHO, 2020), with services concentrated in urban centers. As a result, rural populations, including marginalized occupational groups such as weavers, have very limited access to professional mental health services (Islam and Biswas, 2014). The existing services are predominantly hospital-based and require both travel and expenses that are unaffordable for many low-income workers (Khan, 2020). As a consequence, weavers often rely on informal support systems or traditional healers, who may not have the training or resources to effectively address psychological issues (Haque et al., 2018). From a health systems perspective (Atun et al., 2010), these challenges reflect structural weaknesses such as inadequate financing, workforce maldistribution, a hospital-centric service model and poor integration of mental health into primary care. Addressing these gaps requires embedding mental health within broader health system reforms, expanding community-based services, integrating care into primary health settings and reallocating resources to ensure equitable and accessible support for marginalized rural populations.

### Cultural stigma and help-seeking behavior

Cultural stigma presents another major barrier to accessing mental health care in Bangladesh. Mental illness is often perceived as a spiritual problem, such as possession by evil spirits or divine punishment, rather than being recognized as a medical condition (Hossain et al., 2018). The recent National Mental Health Survey (2018–2019) reported that individuals experiencing mental health issues frequently expressed concern that seeking help from mental health professionals could result in being labeled with stigmatizing terms such as "mad" (WHO, 2019; Hasan et al., 2021). In rural weaving communities, where traditional beliefs are deeply embedded, stigma discourages individuals from seeking professional support and contributes to social isolation. Among women, gendered cultural norms intensify these barriers, as mental distress is often attributed to personal weakness or failure to fulfill domestic responsibilities, which further deters them from pursuing care (Al Azdi et al., 2025). Despite these challenges, cultural frameworks can also function as adaptive resources. Practices rooted in local cultural networks, such as sabr (patience and endurance), family support and social capital, can provide coping mechanisms that support resilience in the face of psychological stress (Griner and Smith, 2006; Lubis et al., 2022; Aggarwal et al., 2023; Islam et al., 2024). Recognizing and integrating such culturally grounded resources into mental health interventions may help reduce stigma and enhance community engagement with mental health services.

## Existing initiatives and promising interventions

Although weavers lack targeted mental health programs, initiatives in related sectors offer adaptable models. The British Asian Trust's Strengthening Mental Health Support for Ready-Made Garment Workers in Bangladesh (2022–2024) trains factory staff as para-counselors and conducts workplace awareness sessions with the goal of reducing stigma and improving mental health literacy (British Asian Trust, 2025). This community-based approach could be adapted for weaving communities by using local cooperatives or workshops as service delivery hubs. Digital platforms also offer promising solutions. The Women Support Initiative Forum, for example, provides anonymous mental health support for Bangladeshi women through online forums and helplines, helping to overcome barriers related to geography and stigma (KolySaba et al., 2022; KolyTasnim et al., 2022; Muhammad and Arafat, 2024). While valuable, these initiatives are not specifically designed for the occupational and cultural context of weavers, which underscores the need for tailored, sector-specific interventions.

Another opportunity lies in integrating mental health into primary care. Bangladesh's Community Clinics have piloted the provision of basic mental health services, including screening and referral (Arafat et al., 2018; Naheed et al., 2022). Expanding this model to meet the specific needs of weavers could substantially improve access to care, although such an expansion would require additional resources, infrastructure and specialized training.

## A novel intervention framework: Community-based, culturally adapted care

This article proposes a pioneering intervention framework aligned with the Double Diamond model (Design Council, 2019), comprising four distinct phases. It begins with a research-informed "discover" and "define" phase, followed by a collaborative "develop"

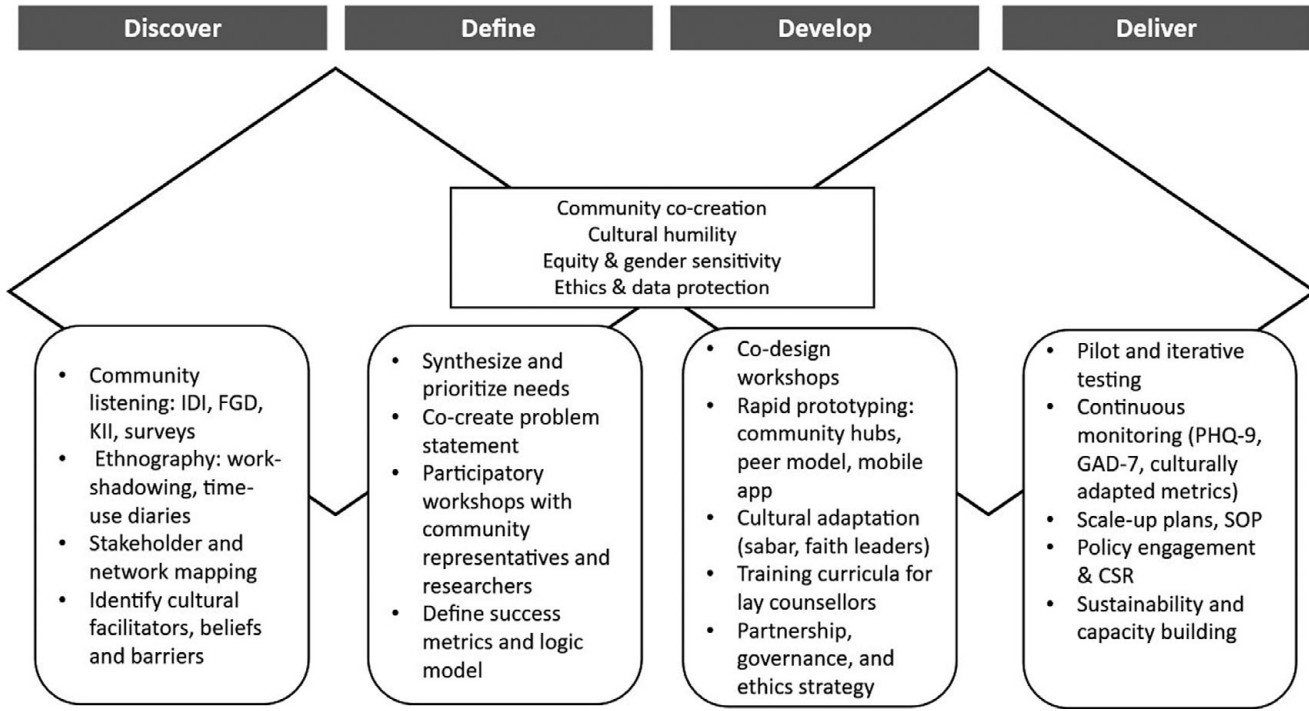

**Figure 1.** Proposed community-based mental health intervention framework for handloom weavers.

and "deliver" phase, to create culturally and contextually adapted interventions. Figure 1 presents the proposed intervention framework for addressing the mental health of weavers (adapted from the Double Diamond model).

### Discover: Exploring the problem space

The "discover" phase focuses on developing a comprehensive understanding of weavers' psychosocial realities. This phase integrates qualitative and quantitative inquiry to identify stressors, stigma and help-seeking behaviors. Community listening sessions, in-depth interviews, focus groups and key informant interviews can reveal factors contributing to mental health issues among weavers, as well as barriers such as stigma and supernatural beliefs, and facilitators such as contextual coping strategies (Lim et al., 2018; Faruk et al., 2023). Ethnographic approaches, including work-shadowing and time-use diaries, may illuminate the embodied strain of weaving, while rapid surveys can provide estimates of symptom prevalence and service utilization. Stakeholder mapping may further identify both formal and informal actors, including religious leaders and traditional healers, who influence mental health decision-making. Importantly, this phase can also highlight protective cultural resources, such as "sabr" (patience and endurance), which may function as adaptive coping mechanisms in the face of adversity (Pargam, 2001; Sony et al., 2022; Tamanna et al., 2023). The outcome of this phase may be a nuanced portrait of the weaving community and a set of preliminary problem statements codeveloped with local stakeholders.

### Define: Problem statement

The "define" phase synthesizes insights from the "discover" phase to create a clear, actionable problem statement. Its primary objective

is to focus on the core challenge that the intervention should address. Participatory synthesis workshops with community representatives and researchers can support the co-construction of problem statements and intervention objectives. By bringing together diverse perspectives, this stage facilitates the identification of feasible, high-impact priorities. Success metrics can be jointly defined to capture clinical outcomes, such as reductions in anxiety and depression, as well as social outcomes, including improvements in trust, reciprocity and social capital (Islam et al., 2024). The logical framework developed at this stage is designed to align local realities with global mental health goals, ensuring coherence between community aspirations and international standards (WHO, 2021a).

### Develop: Codesigning and prototyping solutions

The "development" phase emphasizes creating, refining and adapting prototypes for interventions that are locally relevant and scalable. Codesign workshops can unite weavers, peer leaders, traditional healers and health workers to generate solutions, aligning with evidence that co-creation enhances intervention acceptability and sustainability (Slattery et al., 2020). Prototypes are iteratively refined and adapted to incorporate cultural elements, such as positive religious coping strategies and locally valued practices (Lucchetti and Lucchetti, 2014; Pankowski and Wytrychiewicz-Pankowska, 2023). This framework proposes a multifaceted approach to address mental health challenges among weavers in Bangladesh.

#### Community-based mental health hubs

Establishing community-based mental health hubs within existing local infrastructures, such as weaving cooperatives, union parishad offices or community centers, could significantly improve access to care. These hubs would serve as dedicated spaces for delivering

psychoeducation, basic counseling, group therapy and peer support activities. Integrating mental health services into familiar community spaces not only reduces logistical barriers but also helps normalize discussions around mental health, thereby mitigating stigma. Involving respected local figures, such as union leaders, cooperative heads and religious leaders, could further strengthen community trust. Evidence from related initiatives in Bangladesh shows that culturally embedded, community-owned approaches improve service uptake and sustainability (Faruk, 2022). Training lay counselors within hubs would provide cost-effective and scalable care in settings where professional resources are limited (Patel et al., 2010). These hubs would operate under supervision by psychologists or social workers, with training and minimal technological needs supported by partnerships with NGOs, public–private collaborations and corporate social responsibility (CSR) contributions from textile brands.

### Peer-led support networks

Developing peer-led support networks within weaving communities offers another culturally congruent solution. Trained peers, such as experienced weavers or respected elders, can facilitate group discussions and provide basic psychosocial support. In Bangladesh's collectivist society, people often place greater trust in peers than in external professionals (Koly et al., 2024). These networks can incorporate culturally and religiously relevant coping practices, such as the Islamic concept of sabr (patience and perseverance), storytelling and mindfulness practices adapted to local traditions. Peer supporters can also act as gatekeepers by referring severe cases to formal services. Evidence suggests that peer-led models are cost-effective, scalable and acceptable in low-resource settings (Griner and Smith, 2006). Training, supervision and structured protocols would help ensure quality and sustainability, while digital tools, such as WhatsApp groups, could support coordination. Such initiatives could be funded through government health programs and CSR contributions. Thus, by empowering communities from within, these networks can foster sustained mental health engagement and collective well-being.

### Digital platforms

Digital platforms have the potential to transform mental health service delivery for rural populations by offering scalable, anonymous and culturally sensitive access to psychological support. In Bangladesh, where mental health stigma remains pervasive, especially in rural areas, the discreet nature of digital tools makes them a promising avenue for intervention. Initiatives such as Kaan Pete Roi,[1] the country's first emotional support helpline, has demonstrated success in providing confidential support to individuals in distress, particularly young adults and women who may be hesitant to seek help in person (Iqbal et al., 2021; Jahan et al., 2024). Analysis of calls during its initial 5 years (April 2013–April 2018; $N = 14,344$) revealed that 59.54% of callers were aged 20–39 years, 47.71% were female and the most prevalent concerns involved emotional distress (28.89%), relationship difficulties (27.52%) and financial or educational stressors (13.08%) (Iqbal et al., 2019); these demographic and thematic patterns align closely with those in weaving communities, underscoring the helpline's adaptability to such contexts. Tailored for weavers, digital platforms could be adapted to include self-help

---

[1]Kaan Pete Roi is Bangladesh's first emotional support and suicide prevention helpline, staffed by trained volunteers who offer confidential counseling to individuals experiencing psychological distress.

modules, virtual peer-led support groups and real-time connections with trained para-counselors. These resources would address common barriers, such as geographic isolation, lack of nearby services and stigma associated with help-seeking.

With mobile phone penetration steadily increasing in rural Bangladesh, the feasibility of such digital interventions is rising. According to a recent study in rural Bangladesh, mobile phone ownership has become widespread, even among low-income groups, making digital outreach more inclusive (Khatun et al., 2014). However, successful implementation would require strategic investment in infrastructure (e.g., internet connectivity and device accessibility), user training and collaboration with local stakeholders to ensure digital literacy and cultural appropriateness. When well-integrated, digital platforms can serve as a vital component of a broader, community-centered mental health care system.

### Stakeholder collaboration

In addition to community-based mental health hubs, peer-led support networks and digital platforms, the present framework emphasizes collaboration among government bodies, NGOs, private companies (through CSR programs) and local communities to ensure long-term sustainability and accountability.

### Partnerships for feasibility

To address resource constraints, the framework proposes a multi-pronged funding and implementation strategy. Public–private partnerships and CSR initiatives from international brands can secure financial and technical resources. Linking ethical sourcing with mental health support creates a compelling investment case.

### Governance and ethics strategy

Effective governance is critical for success. The framework proposes establishing a steering committee with representation from local healthcare providers, weaver cooperatives and NGOs. This committee would oversee ethical standards, ensure data privacy and monitor interventions for cultural appropriateness. Table 1 presents the proposed intervention package at a glance.

### Deliver: Implementation and refinement

The "deliver" phase focuses on deploying selected solutions on a limited scale, evaluating their effectiveness and refining them for broader application. The primary objective is to operationalize the framework while ensuring it effectively addresses user needs. A critical step involves piloting community-based hubs, peer-led support networks and digital platforms within targeted weaving communities to facilitate testing and gather feedback. The framework incorporates a robust plan for continuous monitoring to uphold ethical standards and sustain intervention efficacy. This includes employing standardized tools such as the Patient Health Questionnaire-9 (PHQ-9; cutoff ≥10 for moderate–severe depression) (Kroenke et al., 2001), Generalized Anxiety Disorder-7 (GAD-7; cutoff ≥10 for moderate–severe anxiety) (Spitzer et al., 2006) and culturally adapted metrics, with retest intervals of 8–12 weeks to monitor progress. Additional monitoring strategies encompass regular feedback sessions with weavers and ongoing engagement with local stakeholders. Data collected from these efforts will inform iterative refinements, ensuring the

**Table 1.** Proposed intervention package

| Component | Activities | Human resources | Tools | Outputs | Outcomes (example indicators) |
|---|---|---|---|---|---|
| Community-based mental health hubs | Psychoeducation, counseling, group therapy and referral services | Lay counselors, psychologists and local leaders | Training manuals, simple screening (PHQ–9/GAD–7), educational materials (posters and flipcharts), referral forms and community spaces | Increased service access and mental health literacy, stigma reduction, referral of severe cases and community hubs established | Reduction in PHQ–9/GAD–7 scores (≥20% decline), improved community trust, increased help-seeking behavior and reduced stigma |
| Peer-led support networks | Weekly peer support group meetings, storytelling and coping practice sharing (e.g., sabr), referral services and digital group coordination | Trained peers (experienced weavers/ elders), supervisors (community health workers) | Structured peer group protocols, crisis contact lists, WhatsApp/ Viber group for coordination and referral forms | Empowered networks, referrals, increased social support and documentation of local coping mechanisms | Enhanced coping (sabr integration), improved sense of social capital and belonging and increased resilience (measured by Connor–Davidson Resilience Scale) |
| Digital platforms | Anonymous tele-counseling (helplines), virtual peer support, self-help modules (audio/ video) and psychoeducational SMS/IVR messages | Trained tele-counselors/para-counselors, KPR linkage, IT/technical support and content developers (cultural adaptation) | Mobile phones (basic/ smart), IVR/SMS system, apps and helplines (e.g., Kaan Pete Roi) | Anonymous support, helpline availability and culturally adapted self-help modules | Improved access (lower geographical barrier scores), symptom change (PHQ–9/ GAD–7 ≥ 15% decline), user satisfaction, reduced self-stigma and increased mental health awareness |
| Stakeholder collaboration | Funding, oversight and partnerships | Steering committee and NGOs | Governance frameworks | Sustainable resources | Policy integration and long-term funding secured |

intervention remains contextually relevant, effective and sustainable over time (Atun et al., 2010).

## Call to action: Research, policy and collaboration

### Urgent need for research

The mental health of weavers in Bangladesh remains a significantly under-researched area, necessitating immediate and systematic academic attention. Existing literature predominantly focuses on the physical health challenges faced by weavers, such as musculoskeletal disorders, while neglecting the equally pressing psychological burden associated with their occupation (Jamil et al., 2022). Furthermore, most mental health research in Bangladesh focuses on urban populations or specific vulnerable groups like garment workers or students, leaving the weavers, a large, rural and economically disadvantaged workforce, largely invisible in national mental health data (Hossain et al., 2018).

To bridge this gap, mixed-methods research with targeted pilot designs should be prioritized. For instance, a 1-year pilot prevalence survey could be implemented in key weaving districts (e.g., Tangail and Sirajganj), combining quantitative tools like the PHQ-9 for depression and GAD-7 for anxiety (with a cutoff of ≥10 for moderate symptoms, measured at baseline and follow-up, e.g., 8–12 weeks) with qualitative interviews to explore cultural beliefs, stigma, gender dynamics and barriers to care. Longitudinal follow-up studies, tracking cohorts over 2–3 years, would examine how chronic stress and physical pain influence mental health trajectories. Metrics of success could include a 20–30% increase in documented prevalence data accuracy (compared to baseline estimates) and the generation of at least two peer-reviewed publications informing policy. These efforts, funded through partnerships with organizations like the Bangladesh Institute of Development Studies, would provide evidence for culturally tailored interventions while accounting for the sector's informality by engaging informal weaver networks in data collection to ensure representation.

### Policy reforms and workplace regulations

Policy reforms and workplace regulations are critical to reducing the mental health burden among weavers in Bangladesh, where informality, weak labor representation and limited rural governance exacerbate distress. Public investment in mental health remains critically low at 0.44% of the national health budget in the year 2021 (WHO, 2025), far below the 5% advocated by global mental health groups for LMICs (Patel et al., 2018; WHO, 2021b). A phased increase to 2–3% within 5 years is needed, with early priorities including the training of community health workers in mental health first aid to extend services to rural weaving communities. At the labor level, reforms should target excessive hours and low wages by piloting an 8-h workday in cooperative weaving clusters and linking wages to cost-of-living indices. Simplified registration processes would enable informal weavers to access social protections such as insurance and pensions, while capacity-building for cooperatives and unions, supported by NGOs, could strengthen collective bargaining and advocacy. Decentralized monitoring models, in which local unions oversee compliance with the support of district health offices, could improve enforcement in rural areas. Evidence indicates such measures reduce occupational stress and improve well-being (Kabir et al., 2023; WHO, 2024). However, existing frameworks, including the Bangladesh Labour Act (2006, amended 2013) and the Occupational Health and Safety Policy (2013), are weakly enforced and exclude informal sectors, with no specific mental health provisions, leaving most weavers without meaningful protection (ILO, 2021). Therefore, by increasing mental health funding, enforcing labor protections and fostering community-driven oversight, Bangladesh can build a sustainable system that improves well-being and secures the future of the handloom industry.

### Stigma reduction and community engagement

Reducing stigma surrounding mental health is essential to increasing service uptake among weavers, particularly in rural areas of Bangladesh, where traditional beliefs strongly influence health behaviors. Mental illness is often viewed as a sign of spiritual affliction or personal weakness, leading individuals to avoid seeking professional help due to fear of judgment or ostracization (Hossain et al., 2018). Public stigma is further compounded by internalized stigma, which can worsen psychological distress and delay treatment. Community-based stigma reduction campaigns, led by trusted local figures such as religious leaders, village elders and traditional healers, have the potential to transform community perceptions by framing mental health as a shared and manageable concern rather than an individual failing.

Integrating traditional healers into basic mental health training programs, as has been successfully done in other low-resource settings, offers a promising strategy for bridging the gap between cultural and clinical paradigms (Patel et al., 2018). In Bangladesh, where traditional healers are often the first point of contact for mental health concerns, equipping them with fundamental knowledge about mental illness can facilitate timely referrals and enhance community trust in formal care. Such collaborative approaches, grounded in cultural sensitivity and local engagement, are vital for ensuring the long-term success of any mental health intervention in weaving communities.

### Collaboration across sectors

Addressing the mental health needs of weavers in Bangladesh requires coordinated collaboration across multiple sectors. Government agencies must integrate mental health into rural development agendas by allocating resources, implementing policies that support community-based care and expanding mental health services in primary health centers. Such integration aligns with the goals of Bangladesh's National Mental Health Strategic Plan, which emphasizes decentralization and equitable access to mental health services (GOB, 2020).

NGOs, with their deep community reach and programmatic expertise, can play a critical role in scaling mental health interventions tailored to weaving communities. NGOs are well-positioned to deliver culturally adapted psychosocial education, peer-led initiatives and digital health solutions in collaboration with local stakeholders (Doshmangir et al., 2025). Furthermore, international brands that source handloom textiles from Bangladesh have a responsibility to contribute to worker well-being through CSR programs. Ethical sourcing should encompass mental health support, fair labor conditions and investments in sustainable livelihoods.

Establishing public–private partnerships can ensure long-term sustainability of these initiatives. These partnerships can pool financial resources, leverage technical expertise and foster accountability. Nonetheless, a unified advocacy strategy engaging the Ministries of Health and Labour, NGOs and international organizations is essential for integrating weavers' mental health into both national and global policy discourse. Such collective action will be crucial to protecting the mental health of weavers and preserving the future of this heritage industry.

### Reflexivity and limitations

This perspective article is primarily based on secondary literature synthesis and draws on analogies with related labor sectors, particularly garment workers, to infer the mental health risks faced by weavers. While this comparative approach provides a justified entry point, it also introduces limitations regarding generalizability, as weaving communities possess distinct cultural, occupational and gendered dynamics that may not fully align with other informal sectors. A further limitation is the absence of epidemiological data on the prevalence of mental health disorders among weavers, which restricts the ability to quantify burden or design evidence-based interventions. Additionally, the scalability of proposed interventions remains uncertain, given the sector's informality and weak regulatory oversight in rural Bangladesh. Specifically, boundary conditions such as low digital connectivity and weak cooperative density in certain rural clusters could limit the reach of the digital platforms and community hubs, respectively. Nonetheless, acknowledging these gaps is crucial, as they underscore the urgent need for targeted epidemiological studies, mixed-methods research and pilot interventions that can generate context-specific data to inform policy reforms and culturally adapted models of care.

### Conclusion

The mental health of weavers in Bangladesh represents a critical yet largely invisible public health concern, overshadowed by the cultural and economic significance of the industry. Despite employing nearly one million rural workers and contributing over 82 million USD annually to the national economy, the psychological well-being of this workforce remains systematically neglected in both research and policy discourse. The available evidence, although indirect, points to elevated risks of stress, anxiety and depression driven by long working hours, low wages, chronic musculoskeletal pain and existential insecurities linked to mechanization and informality. These vulnerabilities are further exacerbated by pervasive cultural stigma, weak labor protections and gender-based inequities that disproportionately burden women. This article advances a novel intervention framework, guided by the Double Diamond design model, which emphasizes discovery, codesign and iterative delivery. The proposed strategies include community-based mental health hubs, peer-led support networks and digital platforms, all of which are rooted in Bangladesh's collectivist culture and designed to be both scalable and culturally responsive. This framework is transferable to similar LMIC informal-sector contexts, subject to local adaptation and contingent on boundary conditions such as community digital connectivity and cooperative density. While acknowledging gaps in epidemiological data and challenges of scalability, the framework highlights pathways for integrating mental health into broader occupational health, labor rights and rural development agendas. Addressing the mental health needs of weavers is not only a matter of social justice but also a prerequisite for sustaining a heritage industry that is deeply interwoven with Bangladesh's national identity. Urgent, multisectoral collaboration is required to translate these insights into action and ensure that mental health becomes a central component of policies supporting informal labor sectors.

**Open peer review.** To view the open peer review materials for this article, please visit http://doi.org/10.1017/gmh.2025.10081.

**Data availability statement.** The article has used publicly available data.

**Acknowledgements.** The author extends heartfelt gratitude to the weavers of Bangladesh, whose lives, labor and stories inspired this research. Their resilience amid physical hardship, economic uncertainty and social invisibility

serves as the foundation for this work. The author acknowledges their vital contributions to the nation's cultural heritage and economy, often made under challenging conditions. This article is dedicated to amplifying their voices and advocating for their mental health and well-being. Without their lived experiences, this research would not have been possible. The author also extends gratitude to the reviewers for their insightful comments and constructive feedback, which have significantly enhanced the quality and clarity of this manuscript.

**Financial support.** None.

**Competing interests.** The author declares none.

**Ethics statement.** The study did not involve primary data; therefore, ethical approval was not necessary.

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
