## [Reviewer Report]

This perspective piece offers an important contribution by drawing attention to the neglected mental health needs of handloom weavers in Bangladesh, a population often excluded from mainstream policy and mental health discourse. The author demonstrates strong awareness of systemic and cultural factors and proposes a locally grounded, socially inclusive intervention model. The paper aligns with the aims of a perspective article in its conceptual innovation and advocacy tone. Nonetheless, several limitations reduce its effectiveness as a compelling perspective, particularly regarding conceptual grounding, evidentiary scaffolding, and feasibility considerations.

1. Framing and Positioning

One of the essential features of a perspective article is a clearly articulated problem statement, grounded in existing scholarly or policy gaps. While the paper identifies the handloom sector as overlooked, it does not adequately map the state of current evidence. For instance, the author refers to garment sector literature by analogy, but this is not followed by a clear justification for this extrapolation, nor is the scarcity of handloom-specific studies systematically discussed. Consequently, the paper underdelivers on its promise to fill a conceptual void.

Moreover, the perspective does not clearly define its target audience—is it mental health practitioners, public health policymakers, development economists, or cultural theorists? This makes the intended scope of influence ambiguous. A more explicit framing of the paper’s purpose and readership would significantly enhance its strategic orientation.

2. Use of Evidence and Argumentation

Perspective articles are not required to offer new empirical findings, but they must base their claims on a compelling synthesis of literature or field insights. In this manuscript, the literature review is mostly descriptive and lacks critical synthesis. While references to structural stigma, occupational precarity, and physical health burdens are appropriate, the paper does not engage deeply with theoretical models (the social determinants of mental health, syndemics theory, or health systems integration) that could strengthen its conceptual coherence.

There is also a lack of reflection on how cultural beliefs and practices may act as both barriers and resources for mental health adaptation. The treatment of cultural stigma is unidimensional, focusing on supernatural beliefs as barriers, without exploring how local cultural or spiritual networks could be reframed as part of the solution.

3. Innovation and Feasibility of the Intervention Framework

A strong aspect of the paper is its proposal of a novel, community-based, culturally adapted care model. However, the paper does not sufficiently engage with the practical challenges of translating this model into action. Perspective articles must not only propose ideas but also critically assess their implementability. In this regard, the framework lacks analysis in the following areas:

I. Resource feasibility: The human, financial, and technological capacities needed to establish hubs and peer networks are not explored.

II. Governance and integration: There is no discussion of how such interventions would be integrated with the formal healthcare system or monitored for ethical standards.

III. Stakeholder engagement: While the paper promotes bottom-up care, it does not describe any engagement with weavers’ communities, cooperatives, or local mental health organizations in designing the intervention. In a perspective piece, this omission weakens the legitimacy of the proposed model.

4. Forward-Looking Vision and Call to Action

A central goal of perspective articles is to chart a future research or policy agenda. While the manuscript calls for more mental health research and multisectoral collaboration, the recommendations are quite general. The paper misses the opportunity to propose concrete next steps, such as pilot intervention designs, suggested metrics of success, or targeted advocacy strategies.

Additionally, the call to action does not sufficiently account for political economy constraints, such as the informal nature of the handloom sector, weak labor representation, or rural health governance limitations. Without this contextual awareness, the call to action risks appearing aspirational rather than actionable.

5. Authorial Voice and Reflexivity

Perspective papers benefit from a confident yet reflexive tone. While the manuscript is clearly passionate and purposeful, it would be enhanced by some acknowledgment of its limitations, particularly the lack of empirical data, the reliance on analogous sectors, and the assumption of intervention scalability. A brief reflection on these epistemic constraints would elevate the credibility of the author’s position and align the paper more closely with the critical standards expected of high-impact perspectives.

Conclusion

This manuscript engages with a highly relevant topic and has the potential to contribute meaningfully to global conversations on inclusive, culturally grounded mental health strategies. It reflects a commendable effort to imagine grassroots-oriented care structures for marginalized labor groups. However, to realize its full potential as a perspective piece, the article requires greater clarity in problem framing, stronger conceptual grounding, deeper engagement with implementation challenges, and a more detailed roadmap for advocacy and action.

Recommendation: Major revisions recommended, with emphasis on sharpening the argument, contextualizing the intervention model, and deepening the policy analysis.

---

## [Reviewer Report]

I must commend the author for addressing such a neglected topic, which certainly deserves greater attention from both the research community and policymakers.

Introduction:

The introduction requires greater clarity and stronger referencing. For instance:

How many handloom weavers are currently working in Bangladesh?

What is their geographical distribution? What about the indigenous handloom weavers?

Is there any existing evidence regarding the prevalence of mental health issues among this population?

What does the gender or age distribution among handloom weavers look like?

These contextual details should be clearly stated and then linked to national mental health data (e.g., the National Institute of Mental Health [NIMH] survey 2018–2019). This will help build a strong case for prioritising the mental health of this subpopulation.

Mental Health Challenges:

If specific prevalence data on handloom weavers is not available, please avoid drawing unnecessary comparisons with informal workers or other occupational groups unless clearly justified. You can directly acknowledge the absence of data on the mental health of handloom weavers, while noting the evidence available on their physical health. It would also be helpful to briefly discuss the potential links between physical and mental health in this context.

While the subsequent segments appear to make a compelling case, the lack of direct data limits the strength of the argument. I recommend that these sections clearly call for targeted studies to investigate the prevalence and nature of mental health conditions among handloom weavers.

Framework:

As mentioned in the abstract, a framework has been presented—this should be visually depicted using a diagram. Consider aligning the proposed framework with existing design models such as the Double Diamond Framework, and explain its relevance and application.

Overall Structure:

I recommend rewriting the perspective paper with a more direct, structured approach. Clearly outline the rationale for intervention design and propose a pathway that includes:

Cultural and linguistic adaptation of mental health tools

Community-based participatory research

Development or adaptation of context-specific interventions for this unique population

Such a focused and well-structured revision will enhance the credibility and impact of this important piece.

Best wishes and I look forward to read the next version.

---

## [Reviewer Report]

The study is very relevant and important. Highlighting the issues and struggles of handloom weavers in Bangladesh regarding their mental health. I believe this would be a very impactful contribution to the existing knowledge. Here are my critiques:

• The author needs to explain at the beginning of the paper, e.g. in abstract or introduction that how the investigation was conducted Even if it is based on literature or existing research, that should be clearly written as a methodology.

• A brief para on the Weaving industry including number of workers and gender composition would help readers to understand the impact of the topic discussed.

• The author needs to briefly explain ‘Kaan Pete Roi’ at page no. 6, for a reader. Given the word limit it can be in the footnote or in the main body of article.

• Abbreviations should be used across all the paragraphs, for example LMICs were not used in page no. 7

• A short brief about existing labour laws on occupational health and safety would strengthen the call for policy reforms in page 7.

---

## [Reviewer Report]

Dear Author,

Thank you very much for your amendments to the manuscript — very well done. I truly appreciate your hard work.

It was a pleasure reviewing this manuscript.

Best regards.

---

## [Reviewer Report]

Summary verdict

This is a timely and well-argued perspective that usefully foregrounds an understudied workforce. I recommend minor revisions to deepen literature coverage, tighten a few empirical claims, and specify the proposed framework’s operational details.

1) Literature coverage: add key, recent sources and situate claims

Include recent Bangladesh-specific qualitative evidence on female informal workers and migration-related stress. A 2025 phenomenological study on rural-to-urban migrant female garment workers documents pathways from migration stressors to anxiety/depression and coping networks; it offers directly relevant insights for your “informal sector” analogies and gendered analysis.

Anchor the ‘syndemics’ framing in foundational sources. Please cite Singer & Clair (2003) and related updates to clarify definitional criteria (co-occurring conditions, adverse interaction, and harmful social context). This will align your usage with standard public health definitions.

Document the digital/tele-support landscape you reference. When discussing helplines and anonymous support, add an empirical description of Kaan Pete Roi callers and service profile to justify feasibility in weaving districts.

Cite a primary source for the 82.4% musculoskeletal pain figure and briefly describe the sample/context (Sirajganj) to avoid over-generalisation across all weaving clusters. Suggested insertion (end of “Introduction”, after line 96):

“Recent qualitative work on female informal workers in Bangladesh traces migration-related stressors, workplace conditions, and coping strategies that are germane to handloom communities; integrating this evidence would sharpen the gendered pathways we theorise here. We also adopt a syndemics lens in line with Singer & Clair to specify co-occurring burdens and their adverse interactions within harmful social contexts.”

2) Verify and update system-level statistics and attributions

Health-budget share for mental health: You state “0.5%”. The WHO investment case for Bangladesh reports 0.44% (2021); please update the figure and year, and cite the WHO document directly. If you keep “WHO recommends 5%,” attribute carefully (that threshold is widely advocated by global mental-health groups, not a formal WHO numeric target in the Action Plan).

Workforce density: The figure 0.16 psychiatrists per 100,000 is supported by multiple sources; please cite one authoritative source and add the reference year (e.g., Hasan et al. 2021; WHO Special Initiative country profile 2020).

Suggested text tweak (Section “Inadequate mental health infrastructure”):

“Bangladesh allocates approximately 0.44% of the health budget to mental health (2021), and has ~0.16 psychiatrists per 100,000 population, with services concentrated in urban centres.”

3) Clarify scope and inference

Temper generalisations from garment to weaving sectors: Where you infer mental-health burden from adjacent sectors, add one sentence noting contextual differences (production organisation, seasonality, family-based looms) and mark claims as plausible hypotheses pending primary data. This keeps the paper within perspective scope while avoiding over-reach.

4) Gender and equity nuance

Deepen intersectional analysis: You already emphasise women’s double burden; add a line on life-course and marital status (widowed/separated), and briefly consider ethnic minorities (e.g., Manipuri weavers) to align with your district list in the Introduction. This strengthens the equity claim without needing new data.

6) Figures, tables, and placement

Add one short table (“Proposed intervention package at a glance”): Columns: Component | Activities | Human resources | Tools | Outputs | Outcomes (with example indicators). This will help practitioners operationalise your proposal.

7) Language, definitions, and measurement

Define instruments and thresholds where named: When citing PHQ-9/GAD-7 in monitoring, add standard cut-offs and retest interval (e.g., baseline and 8–12 weeks) to make the monitoring plan actionable.

Consistent terminology: Use “handloom weavers (‘tati’)” once, then “weavers” thereafter; standardise “peer-led support networks” vs “peer groups.”

8) Impact statement and claims

Tone down universality: Replace “universally applicable” with “transferable to similar LMIC informal-sector contexts, subject to local adaptation.” Add a clause on boundary conditions (connectivity, cooperative density).